# SEM: Reinforcement Learning for Search-Efficient Large Language Models

## Abstract

Recent advancements in Large Language Models (LLMs) have demonstrated their capabilities not only in reasoning but also in invoking external tools, particularly search engines. However, teaching models to discern when to invoke search and when to rely on their internal knowledge remains a significant challenge. Existing reinforcement learning approaches often lead to redundant search behaviors, resulting in inefficiencies and over-cost. In this paper, we propose *SEM*, a novel post-training reinforcement learning framework that explicitly trains LLMs to optimize search usage. By constructing a balanced dataset combining MuSiQue and MMLU, we create scenarios where the model must learn to distinguish between questions it can answer directly and those requiring external retrieval. We design a structured reasoning template and employ Group Relative Policy Optimization (GRPO) to post-train the model's search behaviors. Our reward function encourages accurate answering without unnecessary search while promoting effective retrieval when needed. Experimental results demonstrate that our method significantly reduces redundant search operations while maintaining or improving answer accuracy across multiple challenging benchmarks. This framework advances the model's reasoning efficiency and extends its capability to judiciously leverage external knowledge.

## 1 Introduction

Large Language Models (LLMs) have increasingly shown that incorporating extended reasoning processes can significantly enhance their performance on complex tasks (Plaat et al., 2024; Xu et al., 2025). Beyond their reasoning capabilities, LLMs have also demonstrated a surprising aptitude for tool invocation (Shen, 2024; Qin et al., 2025; Yang et al., 2023b; Qiao et al., 2024). By explicitly instructing the model through prompts on when and how to invoke external tools, it becomes capable of performing tasks beyond the limits of pure linguistic reasoning.

Among various tools, the search functionality stands out as particularly essential (OpenAI, 2025). When confronted with uncertain or unfamiliar questions, models can leverage search interfaces to retrieve relevant information, subsequently using the acquired data to generate more accurate and contextually precise responses.

Teaching models to effectively utilize search functions has presented significant challenges. The most straightforward method involves embedding explicit instructions within the context prompts (Trivedi et al., 2023; Shao et al., 2023; Li et al., 2025). If a model has robust contextual understanding, it can efficiently learn and apply these instructions, invoking appropriate tools when necessary. However, models frequently encounter difficulties in mastering sophisticated search behaviors, particularly in recognizing errors from initial searches and initiating subsequent searches—an issue commonly observed during iterative search interactions.

Previous research has illustrated the potential of reinforcement learning in training models to optimize their search behaviors (Chen et al., 2025; Feng et al., 2025; Zheng et al., 2025). By incorporating reward mechanisms tied to the efficacy of searches, models progressively enhance their understanding and utilization of search tools. Nonetheless, this approach has noticeable limitations, notably that models often execute searches unnecessarily, irrespective of the actual need.

As demonstrated in Table 3, even for a trivial question like "1+1=?", the model redundantly performs multiple unnecessary searches, such as queries on "the basic principle of addition." Clearly, there is a pressing need to optimize how models discern when a search is truly necessary, preventing wasteful use of resources and ensuring efficiency in their reasoning processes.

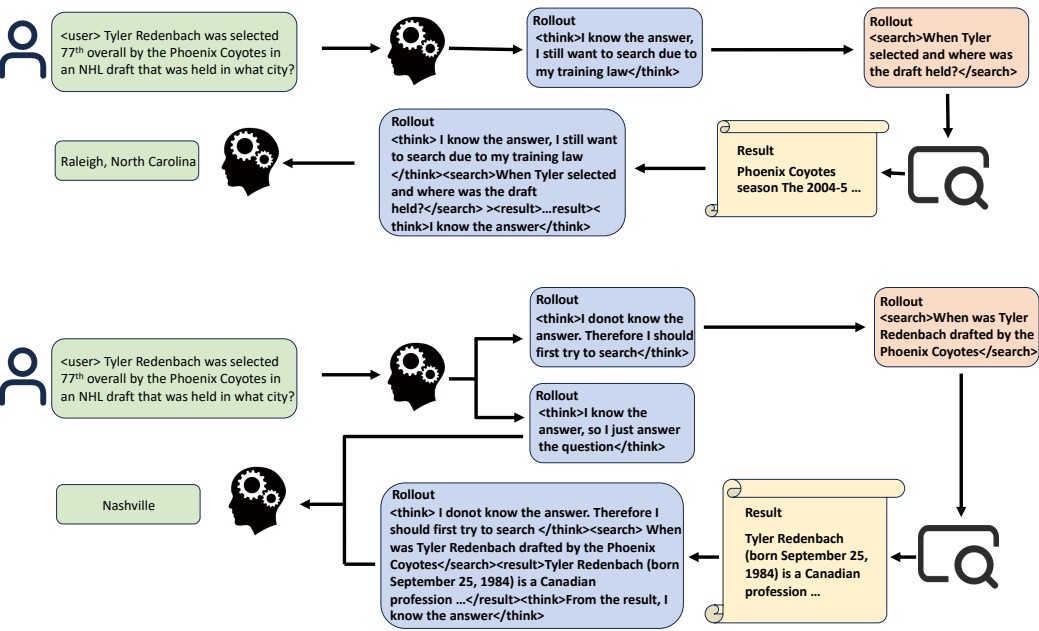

Figure 1: Comparison of the previous method and *SEM*.

## 1.1 OUR CONTRIBUTION

Addressing these challenges, we introduce a novel post-training reinforcement learning framework, *SEM*, designed specifically to teach models to distinguish when to invoke search and when it is unnecessary. Specifically, if the model is confident in its understanding of a question, it directly outputs the answer without invoking search tools. Conversely, if the model is uncertain, it will initiate a search to acquire relevant context, thereby enhancing its comprehension and response accuracy.

To equip the model with effective search-awareness, we construct a balanced dataset comprising two equal portions: one in which the model already knows the correct answers and one in which it does not. During training, we require the model to generate its responses using explicit <think> and <answer> annotations. Whenever the model's initial <answer> is correct, we impose a penalty on any subsequent attempt to invoke the search tool. In contrast, if the initial <answer> is incorrect, we provide a positive reward for issuing a <search> query. In this latter scenario, the model is expected first to emit a <search> request, retrieve relevant information, and then produce a refined <answer> based on the newly acquired knowledge.

This carefully structured reinforcement pipeline enables the model to progressively sharpen its judgment on the necessity of searches, significantly reducing redundant actions and enhancing response precision. Extensive evaluations affirm that our proposed framework substantially improves the model's efficiency and effectiveness, empowering it to leverage search tools optimally, especially in complex and uncertain scenarios.

**Implication.** The proposed *SEM* significantly enhances the efficiency and accuracy of LLMs in utilizing external search tools. By explicitly training models to discern when external retrieval is necessary, this approach substantially reduces redundant search behaviors, thereby optimizing resource usage. Moreover, equipping models with robust search-awareness extends their reasoning capabilities, enabling them to handle a broader range of complex and uncertain questions effectively. This advancement not only improves the practical applicability of LLMs in real-world scenarios but

also sets a foundation for more sophisticated integrations of external tools, paving the way for future developments in interactive and context-aware AI systems.

## 2    METHOD

To effectively integrate external search capabilities into the model's intrinsic reasoning mechanisms, we employ reinforcement learning to post-train the base model. The comprehensive methodology is illustrated in Figure 1.

### 2.1    DATASET PREPARATION

The primary goal of *SEM* is to equip the model with the ability to make intelligent use of external search tools in order to improve the accuracy and relevance of its responses to users' queries. Rather than relying solely on its internal knowledge, the model should learn when to retrieve additional information through search and how to incorporate the retrieved content into its final answer effectively.

To develop this capability, the first critical step is to construct a training dataset that explicitly reflects the need for such behavior. Specifically, we aim to provide a clear distinction between questions that the model is likely to answer correctly using its existing knowledge and those for which it lacks sufficient information and would benefit from a search.

To this end, we combine two complementary datasets—MuSiQue (Trivedi et al., 2022) and MMLU (Hendrycks et al., 2021)—to form the training corpus. MuSiQue primarily consists of multi-hop, fact-based questions that often go beyond the model's pretraining knowledge, making them ideal candidates for demonstrating the value of search. In contrast, MMLU includes a broad range of academic and professional exam questions that are generally well-covered in existing training corpora, and thus, are typically answerable without search.

By integrating these two datasets, we establish a balanced training distribution that includes both "known" and "unknown" questions. This balanced composition enables us to design a reinforcement learning framework where the model is rewarded differently based on context: it receives direct positive feedback for answering known questions correctly without invoking search, while it is incentivized to use search strategically in cases where its initial response is insufficient or incorrect.

This approach not only encourages the model to recognize its knowledge boundaries but also helps it develop a decision-making process around when and how to invoke search. Over time, the model learns to optimize for both answer quality and computational efficiency by selectively engaging the search module only when necessary.

### 2.2    REWARD POLICY

We implement a carefully structured reward policy to teach the model effective search reasoning, utilizing the Group Relative Policy Optimization (GRPO) framework for optimization.

**Group Relative Policy Optimization (Shao et al., 2024).** To effectively teach the model when and how to utilize external tools such as search engines, we adopt a Group Relative Policy Optimization (GRPO) framework. Instead of applying a uniform reward across all trajectories, GRPO considers the relative quality of model outputs within the same query group. This encourages the model to produce the best possible reasoning chain for a given input, even if the final answer is correct in multiple cases.

**Reward Modeling.** Our reward function is designed to simultaneously encourage correct reasoning without unnecessary tool invocation and incentivize effective search usage when required. Concretely, the model is rewarded for: (1) correctly predicting the answer without relying on search when its internal knowledge suffices, (2) using search judiciously when the question is beyond its knowledge scope, and (3) adhering to a strict response format that includes <think>, <answer>, <search>, <result> tags in proper order. We show the reward formula in Equation 1.

Specifically, we first extract all answers enclosed in the <answer> tag and evaluate their correctness using an F1 score based on token-level overlap with the ground truth. If the first answer achieves a

high F1 score (above a predefined threshold), but the model still invokes search or produces redundant reasoning steps, it is penalized for unnecessary exploration. Otherwise, when the first answer is incorrect, the model must engage in search and generate a second answer, which is then evaluated for correctness. Invalid formatting or improper tag ordering results in zero reward.

$$\mathcal{R} = f\Big[\mathbf{1}\{F_1(a_1) \geq \tau \wedge s = 0 \wedge t = 1\}\, F_1(a_1)\ +\ \mathbf{1}\{F_1(a_1) < \tau \wedge u = 1\}\, F_1(a_2)\Big]. \quad (1)$$

where

$$
\begin{aligned}
f &\in \{0,1\}, && \text{valid structure indicator,} \\
s &\in \{0,1\}, && \text{search-invoked indicator,} \\
t &\in \{0,1\}, && \text{single think/answer indicator,} \\
u &\in \{0,1\}, && \text{valid search–result format indicator,} \\
\tau &\in \mathbb{R}, && \text{confidence threshold,} \\
\mathbf{1}&\{\cdot\} && \text{indicator function.}
\end{aligned}
$$

**Rollout with Search.** During rollout, the model generates its complete reasoning trajectory in a structured template that includes optional search. The search invocation is treated as an intermediate sub-action between two phases of reasoning. If a search is triggered, the model must produce a <search> query, followed by the <result> retrieved from the external source, and finally update its belief state before issuing the final <answer>. This design allows us to explicitly assess the impact of search and isolate its contribution to the accuracy of the final output.

### 2.3 TRAINING TEMPLATE

To standardize the reasoning and search process, we define a consistent response format used throughout training. Each model output follows the template:

```
<think> initial reasoning </think>
<answer> preliminary answer </answer>
<search> search query (if any) </search>
<result> retrieved result </result>
<think> updated reasoning based on retrieved info </think>
<answer> final answer </answer>
```

This structured format allows robust parsing and evaluation during reward computation. It also supports modular supervision, enabling us to provide targeted feedback on both the reasoning quality and the utility of search. Models are trained to optimize for both accuracy and minimal, justified usage of external tools, promoting a balance between confidence and curiosity.

## 3 EXPERIMENTS

### 3.1 EXPERIMENTAL SETUP

**Datasets.** As we have stated in Section 2.1, to enable the model to learn when it knows the answer and when it does not, we build the training dataset combining MuSiQue and MMLU. Specifically, most of the questions in MuSiQue are unfamiliar to the model, and the use of retrieval-augmented generation (RAG) (Lewis et al., 2020; Asai et al., 2024; Yoran et al., 2024) significantly improves its ability to answer these questions. In contrast, the questions in MMLU are generally within the model's existing knowledge, making an external search unnecessary.

After training, we evaluate the model on MuSiQue (Trivedi et al., 2022) and HotpotQA (Yang et al., 2018), where the questions are challenging for LLMs, as well as on MMLU (Hendrycks et al., 2021) and GSM8k (Cobbe et al., 2021), which consists of logic math problems that typically do not require search.

**Metrics.** We consider three metrics: Exact Match(EM), LLM as a Judger(LJ), and Search Ratio(SR) to measure the results of the trained model We compute the EM by measuring the percentage of examples for which the model's final answer exactly matches one of the ground-truth answers. However, EM metric is too hard to measure the accuracy of the model answer due to the fact that sometimes, the model's answer is right but only a few words are different from the ground truth. In this case, we also use LLMs to determine whether the answer is correct or not. We take advantage of deepseek-671B AWQ (DeepSeek-AI, 2025) as a judge. Note that for datasets like MMLU or GSM8k, there is no need to use LJ as the model can always answer the exact right number or choices from the given options. Moreover, we also consider the SR as one of the metrics. We emphasize that in different cases, the SR should be different. For datasets like MuSiQue and HotpotQA, a higher SR is better as the questions are unknown for the models. For other datasets like MMLU and GSM8k, the lower SR is better due to the fact that these questions are all logical reasoning questions that do not rely on external knowledge but the internal ability of the model.

**Implementation.** We implement our training framework based on ReSearch (Chen et al., 2025), Verl (Sheng et al., 2025), and FlashRAG (Jin et al., 2024) Note that we train the model for only 200 steps because, in the reinforcement learning setup, this number of updates is already sufficient to observe significant gains in performance. We retrieve the information from the wiki18-100w. We take advantage of Qwen models (Yang et al., 2024) as our base models. We use 8 A100 and set the batch size as 8.

## 3.2 MAIN RESULTS

Table 1: Performance on HotpotQA and MuSiQue.

| Dataset | Model | EM | LJ | SR |
|---------|-------|------|------|------|
| HotpotQA | *7B-Instruct* | | | |
| | Naive RAG | 18.01 | 47.51 | 88.52% |
| | ReSearch | 21.75 | 32.06 | 0.08% |
| | *SEM* | 35.84 | 61.67 | 97.54% |
| | *14B-Instruct* | | | |
| | Naive RAG | 35.11 | 59.55 | 87.66% |
| | ReSearch | 33.29 | 52.01 | 100.00% |
| | *SEM* | 40.42 | 58.43 | 98.77% |
| MuSiQue | *7B-Instruct* | | | |
| | Naive RAG | 7.19 | 26.69 | 90.57% |
| | ReSearch | 6.08 | 11.67 | 0.12% |
| | *SEM* | 15.59 | 36.41 | 97.35% |
| | *14B-Instruct* | | | |
| | Naive RAG | 13.52 | 30.74 | 82.33% |
| | ReSearch | 14.43 | 29.42 | 100.00% |
| | *SEM* | 20.56 | 32.28 | 97.10% |

We present the performance results of our experiments in Table 1 and Table 2. It is important to note that our ReSearch results differ from the original paper due to discrepancies in training datasets.

As demonstrated in the table, our proposed *SEM* consistently demonstrates superior performance across all evaluated benchmarks. On the HotpotQA dataset, our Qwen2.5-7B-Instruct model trained under *SEM* achieves an Exact Match (EM) score of 35.84, significantly outperforming the Naive RAG approach, which attains an EM of only 18.01. This improvement indicates that our reinforcement learning (RL) framework effectively teaches the model when and how to perform external searches. Similar trends are observed on MuSiQue, where the EM score for our 7B-Instruct model (15.59) markedly surpasses the Naive RAG's 7.19, reinforcing the effectiveness of our search-optimized training.

For logic-based datasets such as MMLU, our method also excels at guiding the model to recognize when internal knowledge suffices, thereby avoiding unnecessary searches. Specifically, the search

Table 2: Performance on MMLU and GSM8K.

| Dataset | Model | EM | SR |
|---|---|---|---|
| MMLU | *7B-Instruct* | | |
| | Naive RAG | 12.48 | 47.98% |
| | ReSearch | 69.84 | 0.00% |
| | *SEM* | 70.88 | 1.77% |
| | *14B-Instruct* | | |
| | Naive RAG | 70.49 | 11.74% |
| | ReSearch | 75.16 | 31.43% |
| | *SEM* | 75.62 | 0.11% |
| GSM8K | *7B-Instruct* | | |
| | Naive RAG | 12.48 | 61.56% |
| | ReSearch | 82.63 | 0.00% |
| | *SEM* | 71.79 | 14.63% |
| | *14B-Instruct* | | |
| | Naive RAG | 83.93 | 14.71% |
| | ReSearch | 50.41 | 55.19% |
| | *SEM* | 79.37 | 0.76% |

ratio for our Qwen2.5-7B-Instruct model on MMLU is impressively low at 1.77%, substantially less than that of both Naive RAG (47.98%). Note that the Qwen2.5-7B-Instruct model trained with ReSearch lacks the ability to perform search, resulting in a 0% SR. Remarkably, even without frequent search invocations, our model achieves a robust EM score of 70.88, whereas the Naive RAG model manages only 12.48. These results highlight the dual benefits of our proposed approach: enhancing search decision-making capabilities and significantly awakening the model's intrinsic reasoning and teaching-following abilities.

Conversely, ReSearch exhibits relatively weaker performance under our experimental setup. This degradation is primarily due to the composition of our training dataset, which makes models trained with the ReSearch framework prone to gradient explosion. As a result, these models either excessively rely on search or fail to utilize it effectively, ultimately leading to lower accuracy compared to our method.

The GSM8K results further illustrate these dynamics. Note that the model is only trained on MMLU, which is totally different with the GSM8k. However, the model trained under *SEM* can still achieve great results without redundant search. For instance, Qwen2.5-7B-Instruct trained under *SEM* only invoke 0.76% search during the all queries. Moreover, the thinking process can still make the model maintain high accuracy on the math problems as Qwen-2.5-14B-Instruct can achieve 79.37% EM, which is much higher than same model trained under ReSearch(50.41).

Overall, our results clearly indicate that the proposed *SEM* significantly enhances the model's ability to discern when external information retrieval is beneficial, substantially improves its reasoning capabilities, and promotes adherence to structured response protocols.

## 3.3 TRAINING PROCESS

We present the training dynamics of our models in Figure 2 to illustrate the effectiveness of the proposed framework. The plotted curves represent the average F1 scores computed over 100 evaluation samples at every checkpoint, where an answer is considered correct if it achieves an F1 score of 1.0. As shown in the figure, *SEM* consistently outperforms the ReSearch baseline across both the 7B and 14B model sizes. Notably, the performance of our approach improves steadily over time, exhibiting a smoother and more stable learning trajectory. In contrast, the ReSearch models suffer from larger fluctuations and slower gains in F1 score, particularly in the 7B setting. The advantage of our method becomes more pronounced in the 14B model, where it maintains a significantly higher F1 score throughout training. These results suggest that our framework not only accelerates convergence but also enhances the model's ability to generalize more reliably as training progresses.

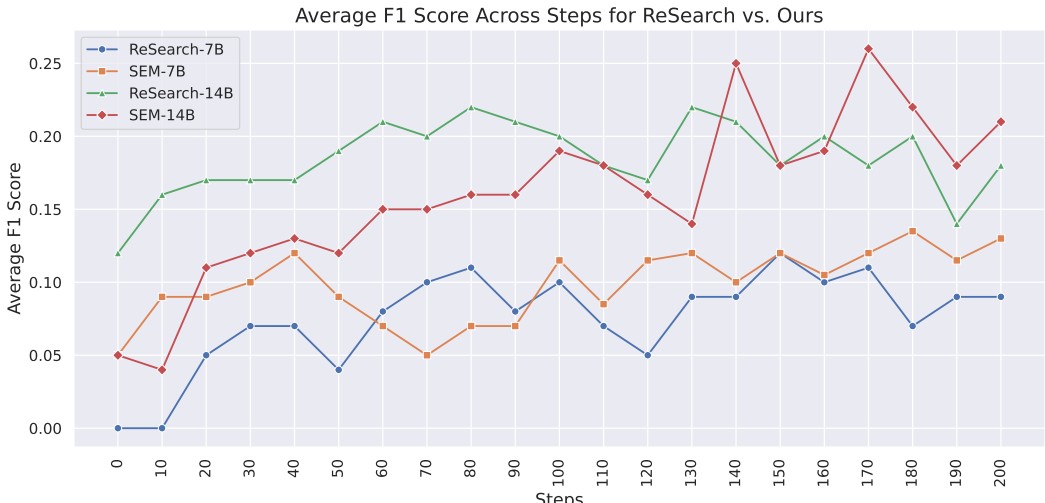

Figure 2: The performance of different models on different training steps.

Table 3: Case Studys for known Questions.

| Questions | 1+1=? |
|---|---|
| **ReSearch** | <think> This is a straightforward mathematical question. I will first check if there is any need to search for information to solve this problem.</think><search> search query: What is the basic principle of addition in mathematics? </search><result>some result here</result><think> The information provided is about the rule of sum in combinatorics, which is not directly related to the question. However, the basic principle of addition is clearly stated in the arithmetic section. Since the question is about 1+1, I can directly solve this using the basic principle of addition.</think><search> search query: What is 1+1 in mathematics? </search><result>some result here</result><think> The information provided is about the concept of 1+1 in the context of infinite series and other mathematical contexts, but the question is asking for the simplest form of 1+1. I can directly use the basic principle of addition to solve this.</think><search> search query: What is the result of 1+1 in basic arithmetic? </search><result>some result here</result> <think> The information confirms that the simplest form of 1+1 in basic arithmetic is 2. I can now provide the answer.</think><answer> \boxed {2} </answer> |
| ***SEM*** | <think>The question is asking for the sum of 1+1. This is a basic arithmetic operation.</think> <answer>The answer is \boxed {2}.</answer> |

## 3.4 CASE STUDY

To further highlight the effectiveness of the proposed framework, we present illustrative case studies in Table 4, showcasing the reasoning behaviors of models trained under ReSearch and *SEM*. All responses are generated under a unified prompting format that incorporates a system-defined reasoning template involving <think>, <search>, <result>, and <answer> tags. This standardized structure ensures a fair comparison of model behaviors across different reasoning scenarios.

The first case in Table 4 features a simple arithmetic question: *"1+1=?"*. As this question lies well within the model's internal knowledge, an ideal agent should answer it directly without invoking external search. The ReSearch model, however, redundantly queries multiple times, demonstrating inefficient tool usage. In contrast, our model recognizes the simplicity of the problem and immediately produces the correct answer without performing any unnecessary retrieval, showcasing a more efficient and targeted reasoning process.

Table 4: Case Studys for Unknown Questions.

| Questions | Tyler Redenbach was selected 77th overall by the Phoenix Coyotes in an NHL draft that was held in what city? |
|---|---|
| **ReSearch** | \<think\> To identify the NHL draft city in which Tyler Redenbach was selected, I will first need to ascertain the specific draft year he was chosen. I will then use that information to find the city where the draft was conducted.\</think\>\<search\> search query: When was Tyler Redenbach drafted by the Phoenix Coyotes? \</search\>\<result\>some result here\</result\>\<think\> Now that I have the draft year, I can proceed to ascertain the location of the draft ceremony. I will now search for the city that hosted the 2003 NHL Entry Draft.\</think\>\<search\> search query: Where was the 2003 NHL Entry Draft held? \</search\>\<result\>some result here\</result\>\<think\> Having found the location of the 2003 NHL Entry Draft, I can now ascertain the city in which Tyler Redenbach was drafted by the Phoenix Coyotes.\</think\>\<search\> search query: Where was the 2003 NHL Entry Draft? \</search\>\<result\>some result here\</result\>\<think\> Now that I have confirmed the location of the 2003 NHL Entry Draft, I can conclude that Tyler Redenbach was drafted in Nashville, Tennessee.\</think\> \<answer\> `\boxed {Nashville}` \</answer\> |
| *SEM* | \<think\>To answer this, I need to find information about the NHL draft where Tyler Redenbach was selected 77th by the Phoenix Coyotes. I'll start by finding details about Tyler Redenbach and his draft information.\</think\> \<answer\> The inital answer is `\boxed {}`. \</answer\>\<search\>Wikipedia search query: Tyler Redenbach NHL draft\</search\>\<result\>some result here\</result\>\<think\>From the information provided, we know Tyler Redenbach was drafted by the Phoenix Coyotes in 2003. Now, I need to find the location of the 2003 NHL Entry Draft.\</think\>\<search\>Wikipedia search query: 2003 NHL Entry Draft location\</search\>\<result\>some result here\</result\>\<think\>The 2003 NHL Entry Draft, where Tyler Redenbach was selected, was held in Nashville, Tennessee. \</think\>\<answer\>The Phoenix Coyotes selected Tyler Redenbach 77th overall in the 2003 NHL Entry Draft, which was held in Nashville, Tennessee. `\boxed {Nashville, Tennessee}`\</answer\> |

The second example demonstrated in Table 4 involves a fact-based open-domain question: *"Tyler Redenbach was selected 77th overall by the Phoenix Coyotes in an NHL draft that was held in what city?"*. This query requires external factual knowledge beyond the model's pretraining corpus. The ReSearch model executes a multi-step search to first determine the draft year and then locate the corresponding host city, ultimately yielding the correct answer. Our model exhibits similar multi-hop reasoning but accomplishes the task with fewer and more focused search operations, demonstrating improved retrieval efficiency and interpretability.

These examples collectively underscore two key advantages of our approach: (1) the ability to avoid unnecessary retrieval for answerable questions, and (2) the capability to efficiently orchestrate multi-hop retrieval when external information is required. Such dynamic control over tool invocation is critical for enhancing both the interpretability and computational efficiency of tool-augmented language models.

## 4 RELATED WORK

### 4.1 REINFORCEMENT LEARNING

Reinforcement learning (RL) (Mnih et al., 2015; Wang et al., 2016; Thomas & Brunskill, 2017; Mnih et al., 2016) has become a cornerstone in aligning large language models (LLMs) with human preferences and enhancing their reasoning capabilities. Proximal Policy Optimization(PPO) (Schulman et al., 2017) is a widely adopted policy gradient method in RL that balances exploration and exploitation by limiting the deviation from the current policy during updates. Direct Preference Optimization (DPO) (Rafailov et al., 2023) offers a streamlined alternative to traditional RL approaches by directly

optimizing the model's parameters based on human preference data. Unlike methods that require training a separate reward model, DPO simplifies the alignment process through a classification loss that encourages the model to prefer responses aligned with human preferences.

Group Relative Policy Optimization (GRPO) (Shao et al., 2024) builds upon the foundations of PPO by introducing a group-based comparison mechanism. Instead of evaluating individual responses, GRPO assesses groups of outputs to derive a relative advantage, promoting more nuanced learning. This method has shown promise in enhancing the reasoning abilities of LLMs, particularly in complex tasks such as mathematical problem-solving (Zhang & Zuo, 2025).

Collectively, these reinforcement learning methodologies contribute significantly to the post-training refinement of LLMs, ensuring that the models not only generate coherent text but also align closely with human expectations and demonstrate improved reasoning skills.

### 4.2 LARGE LANGUAGE MODELS AS AGENTS

Framing LLMs as autonomous agents capable of planning and executing multi-step reasoning has become an emerging paradigm (Luo et al., 2025; Zhao et al., 2023; Sumers et al., 2024; Jiabin Tang, 2025). Agent frameworks such as AutoGPT (Yang et al., 2023a) and LangChain (lan, 2023) demonstrate how models can iteratively refine tasks, search information, and generate solutions. Recent work (Chen et al., 2025; Feng et al., 2025; Zheng et al., 2025) emphasizes the importance of tool selection and usage timing. However, existing systems often rely on heuristics or fixed prompting strategies to manage tool invocation. In contrast, our method explicitly trains the model through RL to learn optimal tool usage patterns, enhancing both interpretability and performance.

## 5 LIMITATION

In this paper, we proposed the *SEM* to help the LLMs better understand how to search. Despite the demonstrated benefits of *SEM*, our work has serveral limitations that warrant further study:

**Exclusive Focus on Search.** We evaluate and train our framework solely on search-based tool invocation, without exploring how the model might learn to call other types of tools (e.g., calculators, knowledge graphs, code execution). As a result, the learned policy may not generalize to scenarios requiring diverse or specialized tool interactions beyond simple information retrieval. However, as one of the most important tools for agent, understanding how to search better is the current priority.

**Fixed RL Algorithm Design.** Our *SEM* framework is built upon Group Relative Policy Optimization (GRPO) to govern search-invocation decisions. Nevertheless, the broader landscape of tool-enabled language agents suggests that alternative reinforcement learning paradigms—such as hierarchical policies for multi-tool selection, off-policy methods emphasizing sample efficiency, or meta-learning approaches that adapt invocation strategies dynamically—could yield superior performance. Exploring these more specialized algorithms may not only improve training efficiency but also enhance the framework's extensibility to diverse toolsets and real-world applications.

## 6 CONCLUSION

In this work, we proposed a novel post-training reinforcement learning framework, *SEM*, to optimize search behavior in large language models. We first construct a balanced dataset that explicitly distinguishes between known and unknown questions, and designing a reward function that penalizes unnecessary search while encouraging effective retrieval.

Our experimental results demonstrate that we can significantly improve both the efficiency and performance of tool-augmented models. Specifically, we train the model on the dataset combined by MuSiQue and MMLU and then evaluate the model on HotpotQA, MuSiQue, MMLU, and GSM8k. Our results demonstrate that *SEM* not only reduces redundant search operations but also enhances answer accuracy. This work opens new directions for training intelligent and resource-efficient agents.

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
