# OpenReview forum: "SEM: Reinforcement Learning For Search-Efficient Large Language Models"
_ICLR.cc/2026/Conference — ICLR 2026 Conference Withdrawn Submission_

### Official Review · Reviewer_GEzQ · 2025-10-28

**Soundness:** 1
**Presentation:** 1
**Contribution:** 2
**Rating:** 0
**Confidence:** 5

**Summary:**

The topic of the paper is interesting. It addresses an important question of how to enable large language models (LLMs) to decide when to act as a research agent that performs external searches versus when to directly answer a question using their internal knowledge.
However, the overall quality of the submission falls below the expected standard. The paper suffers from poor presentation — figures, equations, and templates are presented in a disorganized and visually unappealing way. The evaluation is limited, covering only two datasets and a very small number of baselines, which makes it difficult to assess the claimed effectiveness. Moreover, the novelty is weak, as the approach closely resembles prior work in this area without clear conceptual advancements. Finally, the paper lacks essential implementation details such as training configuration and data statistics, which severely impacts reproducibility. Overall, while the topic itself is promising, the current version of the paper does not meet the bar for publication.

**Strengths:**

1. The topic is interesting.

**Weaknesses:**

1. Poor Presentation Quality
The presentation quality is below the expected standard, giving the impression that the paper was hastily prepared rather than carefully polished. For instance, in Section 2.2 (Reward Formulation), the equation unnecessarily occupies almost half a page, which significantly reduces readability. Similarly, in Section 2.3 (Training Template), the format is presented in raw text, which looks unprofessional and visually unappealing. A structured table or figure would convey the information more clearly and effectively.

2. Limited Evaluation Datasets
The paper can be categorized as developing a research agent, similar to prior work such as [1]. In this line of research, it is standard to evaluate on multiple multi-hop QA datasets (e.g., HotpotQA, MuSiQue, 2Wiki, and Bamboogle). However, the paper only reports results on the first two, which makes the evaluation incomplete and prevents fair comparison with existing work.

3. Insufficient Baseline Models
Given the large number of recent papers on this topic, the choice of only two baseline models is inadequate. A more comprehensive comparison — including recent and competitive baselines — is essential to demonstrate the effectiveness and relevance of the proposed method.

4. Lack of Novelty
The overall novelty of the work is limited. The method appears to follow a similar framework to recent research agents, without introducing substantial conceptual or algorithmic innovations.

5. Missing Implementation Details
The implementation details are very unclear. Key information, such as the learning rate, number of training steps, number of training examples, and data split proportions, is not provided. It seems like the authors don't know we can write unlimited pages in the Appendix.



References:
[1] Huatong Song, Jinhao Jiang, Yingqian Min, Jie Chen, Zhipeng Chen, Wayne Xin Zhao, Lei Fang, Ji-Rong Wen. R1-Searcher: Incentivizing the Search Capability in LLMs via Reinforcement Learning. 2025

**Questions:**

- What are the training details of the framework?

---

### Official Review · Reviewer_Y7Am · 2025-10-31

**Soundness:** 2
**Presentation:** 2
**Contribution:** 2
**Rating:** 2
**Confidence:** 5

**Summary:**

This paper proposes SEM, a post-training reinforcement learning framework for optimizing search behavior in Large Language Models. The authors address the problem of redundant search operations by training models to distinguish between questions they can answer directly using internal knowledge and those that require external retrieval. The approach uses a balanced dataset combining MuSiQue (unknown questions) and MMLU (known questions), and employs Group Relative Policy Optimization (GRPO) to optimize the search behavior.

**Strengths:**

* **Well-Motivated Problem**. The dynamic reasoning/tool-calling issue that this paper aims to solve is a good, practical question.
* **Good performance**. The method shows consistent improvements across multiple benchmarks.

**Weaknesses:**

1. **Limited Technical Novelty**. The paper fails to discuss or compare with established methods that address similar adaptive retrieval problems. Notable omissions include Self-RAG, Adaptive-RAG, and subsequent works that have proposed various solutions for determining when retrieval is necessary. The absence of these discussions makes it difficult to assess the novelty and relative merits of the proposed approach.
2. **Incomplete Experimental Analysis**. For agentic search agents, recent strong baselines such as Search-R1 and related follow-up work are not included in comparisons. For adaptive retrieval methods, Self-RAG, Adaptive-RAG, and similar approaches are absent from the evaluation; The experiments use Qwen2.5 models while the latest Qwen-3 series models are not evaluated.
3. **Missing Ablation Studies**. No ablation experiments are provided to understand the contribution.
4. **Notation and Typography**. The paper contains several typographical errors, particularly in punctuation.

**Questions:**

NA

---

### Official Review · Reviewer_jdqJ · 2025-11-02

**Soundness:** 1
**Presentation:** 2
**Contribution:** 1
**Rating:** 2
**Confidence:** 4

**Summary:**

This paper introduces new post-training framework based on reinforcement learning that encourages the model to adaptively search for questions that require external knowledge and do not search when external knowledge is not needed. To do this, they design a custom rollout style where the model first predicts the response and then based on the initial response decides if search is needed or not. To encourage models to do this, they utilize a custom reward model where encourages the model to first correctly answer the question, then invoke search if the initial response is wrong, and also considers the format. Their experimental results show that they perform well in comparison with the included baselines.

**Strengths:**

- The paper studies an important problem because if they do this successfully, it can decrease the cost of training and inference.
- The reward model designed in this paper is reasonable.

**Weaknesses:**

This paper has a very limited contribution that also doesn’t perform well compared to other studies. For example, [1] achieve much better results on the same datasets without including this adaptive scoring in their reward model. Additionally, many baselines for RAG is not provided that can be further included to make comparison fair. These baselines can be find in [1] paper. Another baseline that studies adaptive retrieval is [2] that should be included. In general, the introduced methods work better than included baselines, but compared to literature (even those with very similar idea) it is not effective.


[1] Search-r1: Training llms to reason and leverage search engines with reinforcement learning
[2] Adaptive-RAG: Learning to Adapt Retrieval-Augmented Large Language Models through Question Complexity

**Questions:**

Why does your method not work as well as [1] or [2] (cited in the weaknesses section) despite using a more complicated reward function that encourages this adaptive retrieval behavior?

---

### Official Review · Reviewer_DMkk · 2025-11-03

**Soundness:** 2
**Presentation:** 2
**Contribution:** 1
**Rating:** 2
**Confidence:** 4

**Summary:**

This paper introduces SEM, a reinforcement learning (RL) framework that train LLMs to efficiently use a search engine to answer open-domain queries. The authors observe that existing models often perform redundant, unnecessary searches despite LLMs can already answer with their embedded knowledge. Therefore, SEM addresses this by training the model on a balanced dataset combining MMLU & MuSiQue, SEM uses GRPO with a reward function that have higher values with correct predictions and less search calls. The experimental results show that SEM can reduces redundant search operations while maintaining or improving answer accuracy across datasets.

**Strengths:**

1. The proposed SEM significantly reduces redundant search operations by training the model to exploit its internal knowledge on simple queries while perform searching when external knowledge is required.

2. The paper introduces a reward function that explicitly penalizes unnecessary searches and rewards effective retrieval, successfully improving search LLMs to distinguish queries it knows and those it doesn't, and only search for those complex ones.

**Weaknesses:**

1. This paper essentially combines reasoning and QA datasets to train a search agent that prioritieze answer correctness followed by the number of searches, with no novel technique / settings introduced.

2. Lack of datasets & baselines. Many baselines like IRCot and Search-R1 are not adopted as baselines in this paper, and further reasoning / QA datasets like MATH & 2Wiki should also be considered to improve the evaluation.

3. Limited training, analysis & observations to demonstrate the efficacy of SEM. For instance, the paper only compares limited baseline methods in Figure 2, whic does not reveal any intrisic advantages / stability of SEM over ReSearch. The authors should also provide more insights / discussions on why SEM may be better than other RL-based search agents and provide potential savings on the API costs.

**Questions:**

N/A

---

### Note · Authors · 2025-11-12

I have read and agree with the venue's withdrawal policy on behalf of myself and my co-authors.